# The Modulation Effect of a Fermented Bee Pollen Postbiotic on Cardiovascular Microbiota and Therapeutic Perspectives

**DOI:** 10.3390/biomedicines11102712

**Published:** 2023-10-05

**Authors:** Laura-Dorina Dinu, Florentina Gatea, Florentina Roaming Israel, Milena Lakicevic, Nebojša Dedović, Emanuel Vamanu

**Affiliations:** 1Faculty of Biotechnology, University of Agricultural Sciences and Veterinary Medicine, 011464 Bucharest, Romania; florentinarom@yahoo.com; 2Centre of Bioanalysis, National Institute for Biological Sciences, 060031 Bucharest, Romania; flori_g_alexia@yahoo.com; 3Faculty of Agriculture, University of Novi Sad, 21000 Novi Sad, Serbia; milena.lakicevic@polj.edu.rs (M.L.); nebojsa.dedovic@polj.uns.ac.rs (N.D.)

**Keywords:** cardiovascular disease, gut dysbiosis, fermented bee pollen postbiotic, microbiota modulation effect, therapeutic strategies

## Abstract

Hypertension is a frequent comorbidity in patients with heart failure; therefore, blood pressure management for these patients is widely recommended in medical guidelines. Bee pollen and postbiotics that contain inactivated probiotic cells and their metabolites have emerged as promising bioactive compounds sources, and their potential role in mitigating cardiovascular (CV) risks is currently being unveiled. Therefore, this preliminary study aimed to investigate the impact of a lactic-fermented bee pollen postbiotic (FBPP) on the CV microbiota via in vitro tests. A new isolated *Lactobacillus* spp. strain from the digestive tract of bees was used to ferment pollen, obtaining liquid and dried atomized caps postbiotics. The modulating effects on a CV microbiota that corresponds to the pathophysiology of hypertension were investigated using microbiological methods and qPCR and correlated with the metabolic profile. Both liquid and dried FBPPs increased the number of the beneficial *Lactobacillus* spp. and *Bifidobacterium* spp. bacteria by up to 2 log/mL, while the opportunistic pathogen *E. coli*, which contributes to CV pathogenesis, decreased by 3 log/mL. The short-chain fatty acid (SCFA) profile revealed a significant increase in lactic (6.386 ± 0.106 g/L) and acetic (4.284 ± 0.017 g/L) acids, both with known antihypertensive effects, and the presence of isovaleric acid, which promotes a healthy gut microbiota. Understanding the impact of the FBPP on gut microbiota could lead to innovative strategies for promoting heart health and preventing cardiovascular diseases.

## 1. Introduction

Cardiovascular (CV) diseases are disorders that affect the heart and blood vessels and have long been the leading cause of mortality worldwide. Heart failure is on the list of the five most common CV diseases, and is caused by alterations in the structure and/or function of the heart. The most prevalent risk factor for the development of heart failure is hypertension, which increases cardiac work. Therefore, medical guidelines recommend rigorous blood pressure management to lower the incidence of heart failure or antihypertensive medication for patients with heart failure, as the prognosis is affected by blood tension [1]. Recently, the complex link between blood pressure and gut microbiota has received considerable attention and some comprehensive reviews have been published [2,3,4,5,6]. Compelling evidence in human and animal models demonstrated the causal bidirectional relationship between hypertension and gut microbiota dysbiosis. The CV microbiota corresponding to the pathophysiology of hypertension shows reduced diversity, increased *Firmicutes*/*Bacteroidetes* ratios, higher abundances of opportunistic pathogens, and reduced populations of acetate/butyrate-producing bacteria [3,6]. However, there is contradictory information about the importance of some microbial taxa in hypertension [4,5]. Targeting gut microbiota to prevent and treat hypertension or to ameliorate the symptoms associated with different CV pathologies is a feasible direction for further adjuvant therapies [7]. 

When thinking about gut microbiota health, the first things that often come to mind are probiotics and prebiotics. However, another player in the field is gaining increasing attention—postbiotics [7,8,9]. Postbiotics are defined by the International Association of Probiotics and Prebiotics (ISAPP) as inactivated microbes and/or their components, including byproducts produced by metabolic activity, which offer a range of potential health benefits [10]. However, related terms, such as “paraprobiotics” and “metabiotics”, have been used in the scientific literature and for commercial products, and at present, there are various scientific opinions on the definition of postbiotics [10,11,12,13,14]. These substances include various metabolites such as short-chain fatty acids (SCFAs), vitamins, enzymes, peptides, and organic acids. The recent interest in postbiotics lies in their stability and safety. Unlike probiotics, which can be fragile and require specific storage conditions, postbiotics can be more easily incorporated into various products, including functional foods, dietary supplements, and pharmaceuticals. Furthermore, as they are non-viable bacterial components, there is no risk of adverse reactions associated with live organisms. While postbiotics are still relatively new, their potential is vast. However, it is important to note that more research is needed to fully understand their mechanisms of action and evaluate their efficacy in different contexts. Regulatory frameworks are also evolving to define postbiotics and establish guidelines for their use.

Healthy dietary products, such as bee products, have been used since ancient times to promote a healthy gut microbiome. Nowadays, fermented bee pollen products such as bee bread are gaining attention, as they are rich in bioactive molecules (e.g., peptides, amino acids, organic acids, vitamins, minerals, and antioxidants) and have better absorption in the human body [15]. Natural or in vitro bee pollen is fermented by specific fructophilic lactic acid bacteria (LAB) strains or other bacteria identified in the bee microflora, increasing the content of bioactive compounds innately found in pollen.

Therefore, this study’s main objective was to investigate the modulation effect of a fermented bee pollen postbiotic (FBPP) on the CV microbiota that corresponds to the pathophysiology of hypertension. A new isolated and identified *Lactobacillus* spp. strain from honeybee microflora was used for lactic fermentation of pollen and to obtain two postbiotic products: liquid and dried atomized caps. This pioneering study shows the changes in CV microbiota composition and metabolic profile after fermented bee pollen postbiotic administration and highlights the therapeutic perspectives of this approach to promote heart health. 

## 2. Materials and Methods

### 2.1. Isolation of Bacteria from Honeybee Microflora and Molecular Identification

In this study, a new *Lactobacillus* spp. strain was isolated from the native gut community of adult worker honeybees (genus Apis) using the method described by Olofsson, 2008 [16]. Stomach samples were placed in 1.5 mL sterile tubes with 1 mL of sterile distilled water and vigorously vortexed. Pure cultures were obtained after a few repetitions of this procedure and subcultured onto De Man, Rogosa and Sharpe (MRS) agar (Oxoid, Thermo Fisher Scientific, Waltham, MA, USA) under microaerophilic conditions at 37 °C for 2–3 days. One colony, presumed to be lactic acid bacteria, was selected, tested for catalase, and Gram-stained, then used for molecular identification. Genomic DNA was extracted with a Quick-DNA Miniprep Plus kit (Zymo Research, Irvine, CA, USA) and the DNA concentration and purity were analyzed [17]. PCR amplification was performed via a Multigene (Labnet International, Edison, NJ, USA) using specific primers for *Lactobacillus* spp.: Lac-1F 5′-ACGAGTAGGGAATCTTCCA-3′ and Lac-2R 5′CACCGCTACACATGGAG-3′ [18]. The PCR product (351 bp) was visualized by gel electrophoresis and confirmed by the melting curve in qPCR reactions. Pure cultures were maintained in MRS broth and glycerol at −80 °C for long-term storage.

### 2.2. Postbiotic Production

The isolated strain was used for anerobic fermentation of culture media based on pollen (provided by Research-Development Institute for Beekeeping, Bucharest, Romania) at 37 °C for 48 h. The medium was obtained by following steps: pollen (10%, g/v) was mixed with 100 mL water and left under agitation for 24 h. The mixture was centrifuged at 4000 rpm for 15 min and glucose 2% was added, similar to the MRS formula. The solution was sterilized at 110 °C for 5 min. A fresh culture of *Lactobacillus* spp. isolate was used (1%) to inoculate the sterilized media based on pollen. To obtain the liquid FBPP, the microbial suspension obtained at the end of fermentation was thermally treated at 60 °C for 10 min. To obtain the dried FBPP, the microbial suspension was supplemented with 7% calcium fructoborate and spray-dried. The dried powder was used to fill a type 00 gastro-resistant capsule. Both postbiotic preparations were plated onto MRS agar media and non-selective media Nutrient agar (Oxoid, Thermo Fisher Scientific, Waltham, MA, USA) to confirm the absence of viable cells.

### 2.3. In Vitro Simulation Using the GIS1 System

Tests were conducted using a gastrointestinal simulator (a GIS1 system), only in phase 2 (the human colon) (http://www.gissystems.ro/, accessed on 1 August 2023) and the microbiome of the target CV disease—the dyslipidemia and hypertension group. The reconstitution process followed the protocol previously described and was performed with a mean interval of 7–10 days [19]. The samples (feces) were handled by the ethical guidelines of UASVM Bucharest (ColHumB Registration number: 1418/23 November 2017; www.colhumb.com) and analyzed individually. The biological samples were collected in 10% glycerol and stored at −15 °C until needed. The samples used in this study were obtained from individuals clinically diagnosed with cardiovascular diseases, including hypertension and dyslipidemia. It is important to note that these individuals did not use any medications that could potentially impact the pattern of their microbiota, such as antibiotics. The collecting techniques for fecal samples adhered to the ethical guidelines set forth by the University of Agricultural Sciences and Veterinary Medicine in Bucharest, Romania. Three individuals (two men and one female) aged 45 and 70 were selected to collect fecal samples. The frozen glycerol samples were reconstituted in phosphate-buffered saline (PBS) to achieve a consistent microbial profile. The liquid and capsules with FBPP were directly added to the simulated environment under sterile conditions. The capsules used were gastro-resistant and could pass through phase 1 of the simulation (stomach and small intestine). The pH evolution was recorded during the simulation process. At the end of the in vitro simulation (day 10), each sample collected was subjected to centrifugation at 4000× *g* for 15 min (Hettich Universal 320, Hettich GmbH., Kirchlengern, Germany), and the sediment (microbial fingerprint) was microbiologically analyzed within 24 h or preserved in glycerol 20% for qPCR analysis. The supernatant was stored in a refrigerator at −15 °C for chemical analysis. Two different experiments were performed.

### 2.4. Microbiome Analysis Using Microbiological Methods and the qPCR Technique

For microbiological analysis, samples were diluted and plated from appropriate dilutions onto selective media: MRS agar (Oxoid, Thermo Fisher Scientific, Waltham, MA, USA) for *Lactobacillus* spp., BSM agar (Sigma-Aldrich, St. Louis, MO, USA) for *Bifidobacterium*, MacConkey agar (Oxoid, Thermo Fisher Scientific, Waltham, MA, USA) for *Enterobacteriaceae*, and non-selective media Nutrient agar (Oxoid, Thermo Fisher Scientific, Waltham, MA, USA).

Genomic DNA from samples was isolated using the Quick-DNA Miniprep Plus kit (Zymo Research, Irvine, CA, USA) according to the manufacturer’s instructions for bacteria, and quantified with a NanoDrop 8000 (Thermo Fisher Scientific, Waltham, MA, USA) to assess DNA concentration and purity. All qPCR reactions were performed using a Rotor-Gene 6000 5plex HRM (Qiagen-Corbett Life Science, Sydney, Australia) instrument and software to generate the standard curve and microbial quantification. Standard curves were routinely performed for each qPCR using serial dilutions of control DNA from *E. coli* ATCC 8739 (primer set uivA-7F/uivA-7Rdeg) for *Enterobacteriaceae* quantification (R^2^ = 0.9953). A mix of *Lactobacillus acidophilus*, *L. plantarum* and *L. rhamnosus* for *Lactobacillus* sp. (primer set Lac-1F/Lac-2R) were used for *Lactobacillus* spp. and *Firmicutes* (primer set Firm-934F/Firm-1060R) quantification (R^2^ = 0.9918). A mix of *Bifidobacterium animalis* and *B. bifidum* were used for *Bifidobacterium* sp. (primer set g-BIFID-F/g-BIFID-R) quantification (R^2^ = 0.9995). A strain of *Prevotella* sp. was used for *Bacteroidetes* phylum (primer set Bac-960F/Bac-1100R) quantification (R^2^ = 0.9682) [18,20,21]. The final volume in all reactions was 25 μL, including 1 μL of template DNA, 12.5 μL of Maxima SYBR Green Mix (Thermo Fisher Scientific, USA) and 0.5 μL of each primer. The cycling parameters for the 3-step melting process were 10 min at 95 °C, followed by 40 cycles for 15 s at 95 °C, 30 s at 60 °C, and or 45 s at 72 °C, based on the PCR product size. The specificity of real-time PCR reactions was confirmed by melting curve analysis. Reactions were carried out in triplicate, and the results were statistically analyzed.

### 2.5. Analysis of Organic Acids Produced by the Fermentation Process to Obtain Postbiotics and after in Vitro Simulations

A zonal electrophoretic method with reverse polarity was used, similar to previous studies [22,23]. All used reagents were of analytical purity (purity > 98%): D,L-lactic and butyric acids were purchased from Fluka (Buchs, Switzerland); acetic acid was purchased from Riedel-de-Haën (Seelze, Germany); and formic, benzoic, succinic, 3-(-4-hydroxyphenyl) lactic, phenyllactic, isovaleric and propionic acids were purchased from Sigma-Aldrich (St. Louis, MO, USA). Phosphoric acid 85% and oxalic acid were purchased from Merck (Darmstadt, Germany), cetyltrimethylammonium bromide (CTAB) from Loba Chemie (Fischamend, Austria), and chromatographic purity water, NaOH 0.1 N and 1 N from Agilent Technologies (Santa Clara, CA, USA). SCFAs separation was performed using Agilent G7100 capillary electrophoresis apparatus (Agilent Technologies, Ratingen, Germany) with a diode array detector (DAD). A standard silica capillary with a diameter of 50 m and an effective length of 63 cm was used for the separation. The migration buffer, adjusted to a pH of 6.24, comprised H_3_PO_4_ 0.5 M (Cetrimonium bromide), CTAB 0.5 mM (pH adjusted with NaOH to 6.24), and 15% methanol. The separation occurred at 25 °C and a voltage of 20 Kv. Samples were injected hydrodynamically for 10 s at 35 mbar. Detection was performed on a DAD at a wavelength of 200 nm. Between sample migrations, the capillary was washed for 2 min with 1 M NaOH, 2 min with ultrapure water, and 3 min with the BGE. Standard stock solutions were prepared in water and kept at +4 °C. Dilutions of the common solutions were made daily. The analysis time for each run was 30 min. The separated compounds were identified by comparing the retention times and using standard addition. Before injection, all the samples were filtered (using 0.2 m diameter membranes from Millipore, Burlington, MA, USA) and degassed.

### 2.6. Statistical Analysis

All the parameters were evaluated in triplicate, and the results were expressed as means and standard deviation (SD). Statistical analysis was conducted using the IBM SPSS Statistics 23 software package (IBM Corporation, Armonk, NY, USA). One-way and two-way ANOVAs were used for metabolomic activity and gut microbiota analyses, followed by Dunnett’s test. The significance level for the calculations was set as follows: significant, *p* < 0.05; very significant, *p* < 0.01; highly significant, *p* < 0.001; and extremely significant *p* < 0.0001 using letters from a to d. 

## 3. Results

### 3.1. Fermented Bee Pollen Postbiotic Production

Following the isolation protocol, four isolates were obtained showing phenotypic character similar to *Lactobacillus* species on species-specific MRS agar media, but with slower growth rates. All the isolates were found to be rod-shaped, non-spore-forming, Gram-positive bacteria and catalase negative. The selected isolate for further experiments was identified using a specific primer set (Lac-1F/Lac-2R), previously developed by Rinttilä et al. in 2004 for targeting the 16S rDNA gene region [18].

The *Lactobacillus* spp. bee isolate was used for anerobic fermentation of culture media containing 10% pollen collected from Romania and 2% glucose. Throughout the fermentation process, bacteria disintegrated the multilayer wall of the pollen grain, including the inner coating that contains pectin and cellulose, while enhancing the pollen compounds’ bioavailability (minerals, vitamins, and polyphenolic compounds, mainly phenolic acids and flavonoids). Moreover, during natural or in vitro pollen fermentation, bioactive molecules, such as amino acids, peptides, enzymes, and organic acids, are produced [15]. After 48 h of fermentation, the microbial suspension was heated (60 °C) or spray-dried to obtain non-viable bacteria and to preserve the bioactive substances. Table 1 shows the organic acid composition of the microbial metabolite from dried FBPP, which was analyzed using a zonal capillary electrophoresis method (Table 1). 

The composition of the postbiotic suspensions is complex and eight organic acids were detected in total, in addition to an appreciable amount of lactic acid (18.671 ± 0.429 mg/mL) and acetic acid (2.237 ± 0.027 mg/mL). A major characteristic of LAB strains is the capacity to produce lactic and acetic acids as the principal or only fermentation product. Other active substances that LAB strains produce are phenyllactic acid and hydroxyphenyl lactic acid, both detected at the lowest level. Other SCFAs, i.e., butyric and propionic acids, were identified alongside small amounts of 3-methylbutanoic acid, also named isovaleric acid.

### 3.2. The Effect of Fermented Bee Pollen Postbiotic on the CV Microbiota Composition

Changes in gut microbiota composition have been linked to CV pathologies, such as atherosclerosis, hypertension, heart failure, or coronary artery disease [2,3]. The CV gut microbiota used in the study has been previously analyzed and used in different studies [24,25]. This microbiota corresponds to the pathophysiology of hypertension with an increased *Firmicutes*/*Bacteroidetes* ratio (*F*/*B* ratio = 4.03 in the control sample), which is generally considered a signature of CV gut dysbiosis. The *F*/*B* ratio decreased to 3.77 in samples incubated for a few days with liquid FBPP, but the value did not statistically significantly change (Figure 1). The two major phyla *Firmicutes* and *Bacteroidetes* encompass almost 90% of microbial species inhabiting the human gut, and an increased *F*/*B* ratio has been associated with CV diseases, obesity, diabetes, and inflammatory bowel disease, as well as with different dietary structures [6,26]. 

Subsequently, CV dysbiosis is linked to reduced levels of microbial diversity, a loss of beneficial taxa, and a decrease in bacterial loads [6,20]. Our data showed that total bacterial counts in samples treated with postbiotics increased. Thus, in the sample treated with liquid FBPP, the total number of culturable germs, both aerobes and anaerobes (log 17.655 CFU/mL), is 5.2 log CFU/mL higher than in the control sample (log 12.431 CFU/mL)—Figure 2. 

Both samples with postbiotics showed an increased number of *Lactobacillus* spp. based on the qPCR results, while the amount of culturable bacteria in MRS media for the sample treated with liquid FBPP is approximately 2 log CFU/mL higher. In samples treated with spray-dried FBPP, the lactobacilli density measured by the qPCR technique was 9.675 ± 0.132 log/mL, while for liquid FBPP, the lactic acid bacteria population was 10.350 ± 0.057 log/mL. Another beneficial bacterium, acetate-producing *Bifidobacterium* spp., increased in dried and liquid FBPP, based on both detection methods, qPCR, and cultivation in selective BSM media. The number of bifidobacteria exceeded 8.750 ± 0.602 log/mL in samples treated with liquid postbiotic and was log 7.40 ± 0.450 log/mL in the dried postbiotic sample, compared to 7.050 ± 0.458 log/mL in the control samples (Figure 1). A depletion in beneficial bacteria, including SCFAs producing *Lactobacillus* spp. and *Bifidobacterium* spp., has been reported in many disease conditions and is recognized as a feature of CV dysbiosis [20,27]. 

Concurrently, an increased amount of *Enterobacteriales* was associated with a rise in systemic inflammation and intestinal permeability, both associated with hypertension [27]. Using qPCR detection samples treated with postbiotic showed that *E. coli* significantly decreased (up to 3 log CFU/mL) compared to the control sample and similar data were obtained with culturable bacteria in MacConkey media (Figure 2).

### 3.3. The Effect of Fermented Bee Pollen Postbiotic on the Metabolomic Profile of CV Dysbiosis

Table 2 shows the organic acid profiles of control and FBPP-treated samples. The metabolic profile after the gastrointestinal simulation (GIS) process is correlated with the changes in microbial composition, including lactate and acetate production. 

Minor differences between results obtained after administration of liquid and spray-dried postbiotics could be correlated with the changes in the postbiotic composition after the atomization process. Before FBPP treatment, small amounts of lactic acid (0.39 ± 0.05 g/L), acetic acid (0.34 ± 0.04 g/L), and propionic acid (0.018 ± 0.01) were detected, similar to previously published data [25]. 

Both techniques, zonal capillary electrophoresis and HPLC found high concentrations of acetic acid in samples treated with fermented bee pollen postbiotic suspensions. Thus, acetate was also detected at a lower level in postbiotic preparations before simulation (2.237 ± 0.027 mg/mL) and the amount doubled after gastrointestinal simulation in samples treated with dried FBPP (4.284 ± 0.017 g/L), while the control sample contained 3.083 ± 0.030 g/L. Similarly, lactic acid was identified before and after GIS, with the highest concentration in samples treated with spray-dried postbiotic (6.386 ± 0.106 g/L) and liquid postbiotic (5.435 ± 0.131 g/L), compared to the control sample (0.701 ± 0.020 g/L). Moreover, small quantities of phenyllactic and hydroxyphenyl lactic acids were detected before, and these increased after in vitro stimulation with FBPP. Another microbial metabolite SCFA detected at low levels before and after postbiotic administration was butyric acid, which was not detected in the control sample. Propionic acid was intensively produced by some microbial strains in the control sample but not in samples treated with fermented bee pollen postbiotic. 

Notably, benzoic acid needs to be added to samples after FBPP administration. However, isovaleric acid was detected in samples treated with liquid postbiotic (2.096 ± 0.016 g/L) and spray-fried postbiotic (2.725 ± 0.037 g/L) in a concentration 4–5 times higher than the control sample (0.544 ± 0.008 g/L).

## 4. Discussion

Postbiotics represent an exciting frontier in gut health research. The metabolites derived from probiotic bacteria have the potential to provide a range of health benefits, from supporting the gut function to influencing immune and metabolic health. As our understanding of postbiotics deepens, we can anticipate the development of innovative strategies to harness this therapeutic potential and promote overall well-being.

In this research, a new isolated strain from the bee gut community and identified as *Lactobacillus* spp. was used for lactic fermentation of pollen to obtain two postbiotic products: liquid and dried atomized caps. Natural or in vitro lactic fermentation of bee pollen increases the content of bioactive compounds in fermented products because of enzymatic transformation and the release of biologically active substances consistently found in bee pollen, such as vitamins and minerals [15]. Through fermentation by probiotic bacteria, pollen’s rich and complex nutrient content is transformed into beneficial metabolites (SCFAs, enzymes, peptides, and polyphenolic compounds) that enhance the quality of the product and increase bioavailability [15,28]. Therefore, using bee pollen as a raw material for postbiotic production offers health benefits and presents a sustainable and environmentally friendly approach. Pollen is a naturally abundant and renewable resource, making it an attractive option for obtaining postbiotics on a larger scale.

A total of 11 organic acids were analyzed in dried fermented bee pollen postbiotics but only eight compounds were detected, which did not include formic, benzoic, and succinic acids. The main microbial metabolite of LAB bacteria was found at the highest level compared to other organic acids—18.671 ± 0.429 mg/mL lactate. Additionally, acetate (2.237 ± 0.027 mg/mL) was found in higher concentrations compared to propionate (0.051 ± 0.002 mg/mL) and butyrate (1.066 ± 0.016 mg/mL). Using HPLC-DAD analysis, a recent study on Romanian bee bread demonstrated the presence of the following organic acids: gluconic acid (79.2 g/kg), formic acid (6.75 g/kg), acetic acid (10.7 g/kg), propionic acid (1.3 g/kg), and butyric acid (0.33 g/kg) [29]. Moreover, lactic acid (0.72–1.2 g/kg) was detected in Anzer and chestnut pollens by Kalaycioglu et al., 2017, which confirmed the antimicrobial properties of the organic acids from pollen and fermented bee bread [30]. Knazovická et al., 2018, using a natural fermentation model of bee pollen demonstrated that the final product, named pollen can, has a lower pH (approximately 4.5), caused by lactic and acetic fermentation that ensure the spoilage safety [28]. Other studies found improvements in the nutrient content of the resulting products from bee pollen fermentation with selected LAB strains, including the total phenolic and flavonoid contents [31,32]. However, fewer studies have been published on the organic acid composition of pollen or bee bread obtained by natural or in vitro fermentation. Results are difficult to compare as the nutritional composition varies based on the pollen’s local and seasonal value and the fermentation protocol influences the quality of the final products [15]. Besides their chemical composition, postbiotics exhibit enhanced safety and stability, enabling their application in the food and pharma industries. 

Hypertension is the main risk factor for many CV pathologies, including heart failure. Therefore, this preliminary work investigated the in vitro modulation effect of fermented bee pollen postbiotic on CV microbiota associated with hypertension, and the results on changes in the microbial composition were compared with data of the metabolic profile after GI simulation. SCFAs, such as acetate, butyrate, and propionate, are significant microbial metabolites extensively studied for their positive effects on gut microbiota health. They provide energy for intestinal cells, possess anti-inflammatory properties, and help maintain a healthy gut barrier function [33]. Animal studies have demonstrated that a reduction in SCFA-producing bacteria is related to increased blood pressure [34,35,36]. More research is required in humans to understand the antihypertensive effects of SCFAs [4,5,37]. However, acetate and propionate are suggested to lower blood pressure by decreasing systemic inflammation, while lactate and butyrate impact blood pressure through vasodilatation/vasoconstriction mediated by distinct G-protein coupled receptors [2]. 

By fermenting pollen, *Lactobacillus* spp. produced SCFAs, contributing to the overall pool of beneficial molecules in the gut. Acetic acid, one of the major SCFAs, plays a crucial role in microbiota modulation and has several important functions in the context of the gut microbiota. It acts as an energy source for colonic epithelial cells and helps regulate the pH (acidity) of the gut environment [38]. This acidic environment is unfavorable for certain harmful bacteria but promotes the growth of beneficial bacteria, such as *Bifidobacteriaceae*. Additionally, acetate has been shown to possess anti-inflammatory properties and helps strengthen the intestinal barrier, which is the gut’s protective lining [39]. A healthy gut barrier prevents the translocation of harmful substances or pathogens from the gut into the bloodstream, maintaining gut health. In our study, acetic acid was detected at a lower level in postbiotic preparation before simulation (2.237 ± 0.027 mg/mL) and was produced during gastrointestinal simulation, especially in samples treated with dried FBPP (4.284 ± 0.017 g/L) and less so in untreated CV microbiota (3.083 ± 0.030 g/L). At the same time, up to 1.7 log/mL increases in the populations of acetate-producing *Bifidobacterium* spp. were detected by qPCR after cultivation in specific media and are associated with a higher concentration of acetic acid in both samples after postbiotic administration. Reports have consistently revealed that *Lactobacillus* abundance is linked to blood pressure [2,3,5]. Increasing *Lactobacillus* spp. with long-term probiotic treatment or reducing salt intake, which depletes these bacteria, has a blood pressure attenuation effect [5]. Additionally, it has been suggested that increased nitric oxide production may play a critical role in the antihypertensive effect of *Lactobacillus* spp. [36]. Therefore, numerous studies have demonstrated the potential of *Lactobacillus* spp. and *Bifidobacterium* spp. for regulating blood pressure, showing that certain species are enriched in healthy or treated groups to lower blood tension [4]. In our study, a higher *Lactobacillus* spp. density in postbiotic-treated samples was correlated with the lactic acid concentration. In samples treated with spray-dried FBPP, the lactobacilli density determined after cultivation in MRS media was 7.759 ± 1.187 log CFU/mL and the lactate was 6.386 ± 0.106 g/L, while for liquid FBPP, the lactic acid bacteria population was 9.117 ± 0.525 log CFU/mL and the lactate was 5.435 ± 0.131 g/L. Higher numbers of lactic bacteria detected by qPCR in samples treated with FBPP could be linked with the presence of DNA from honeybee *Lactobacillus* inoculum, but the increased amount of culturable lactobacilli in specific media compared to the control is entirely the effect of the liquid postbiotic. 

Moreover, the acidic environment produced by acetate and lactate and the increased beneficial taxa could explain the significant decrease in opportunistic *Escherichia coli* (up to 3 log CFU/mL) in both samples treated with postbiotic compared to the control sample. Diets rich in fiber that increase *Bifidobacteriaceae* are known to be protective against pathogenic bacteria and subsequently lower blood pressure. A low pH in fermented bee pollen products decreased *Enterobacteriaceae* amounts in these products compared to bee pollen [28]. Disbiotic patterns in patients with a high blood pressure are reported alongside a greater abundance of opportunistic Gram-positive and Gram-negative pathogens (e.g., *Escherichia coli*, *Klebsiella* spp., and *Streptococcus* spp.), while the extended abundance of *Escherichia* and *Shigella* taxa seems to contribute to CV pathogenesis [3,6,27]. 

Another well-known SCFA is butyrate, which acts as a source of energy for cells lining the colon, supports a healthy gut barrier function, and possesses anti-inflammatory properties [37,40]. This organic acid was detected at low levels before and after postbiotic administration but not in the control sample. Reduced populations of butyrate- and acetate-producing bacteria were reported as a biomarker for CV dysbiosis in human and murine models [20]. Remarkably, benzoic acid is missing in samples after fermented bee pollen postbiotic administration compared to the control sample (0.061 ± 0.006 g/L). This last result is important because benzoic acid can reach the gut, where it may interact with the resident gut microbiota. Studies have shown that high levels of benzoic acid can adversely affect the diversity and composition of the gut microbiota [41]. It may selectively inhibit the growth of certain beneficial bacteria while allowing the proliferation of less desirable species. In our study, isovaleric acid was produced by certain bacteria in the gut microbiota through the fermentation of the main substrate (pollen), particularly those containing amino acids such as leucine, isoleucine, and valine. In samples treated with FBPP, the concentration of isovaleric acid was 4–5 times higher than in samples of untreated CV microbiota. Some gut species of *Propionibacterium* are known to produce isovaleric acid. *Propionibacterium freudenreichii* CIRM-BIA 129 is recognized as a potential probiotic because of its ability to confer health benefits, especially anti-inflammatory effects [42]. Additionally, bacteria can survive and colonize the human gastrointestinal tract, promoting a healthy gut microbiota [43]. 

The ratio of *Firmicutes* to *Bacteroidetes* (*F*/*B*) has been a subject of interest in research related to CV pathogenesis, obesity, and metabolic health. Some studies have suggested that obese individuals may have a different *F*/*B* ratio in their gut microbiota than lean individuals [43]. This observation has led to the hypothesis that a higher *Firmicutes* to *Bacteroidetes* ratio might be associated with increased energy extraction from the diet, greater fat storage, and an increased risk of obesity. In the case of cardiovascular pathologies, an increased *F*/*B* ratio is considered a biomarker for CV dysbiosis [3,6]. In this study, no statistically significant changes in the *F*/*B* ratio were observed, probably because of the short-term administration of the treatments (10 days). Thus, the spray-dried administration of the postbiotic determined a higher *F*/*B* ratio (4.9) than liquid administration, which had a lower ratio of 3.77, approximately 20% lower than the control. Moreover, the total bacterial load in samples treated with postbiotic increased. 

In recent years, studies to prove the therapeutical potential of strains isolated from bee bread or used in the fermentation process of bee pollen have received considerable attention [15]. Bee bread has been suggested to have a prebiotic effect based on the native bioactive compounds in pollen and the newly produced molecules, especially proteins, polyphenols, vitamins, minerals, and organic acids [15,31,44,45,46]. Moreover, bee bread and fermented bee pollen products are probiotic products with consequent health benefits and longer shelf lives [15]. However, only a few reports have been published on the functional potential of fermented bee products in the intestinal microbiota using human, mouse, and zebrafish models [31,47,48]. Recently, the cardioprotective potential of various bee products has been reviewed and it was concluded that these products may be effective in the prophylaxis and treatment of cardiovascular diseases; however, more studies are required, especially to improve the knowledge gaps regarding the cardioprotective mechanism [49]. Major royal jelly protein, bee pollen, and propolis have been proven to have an anti-hypertension effect in murine models [50,51]. Our research attempts to shed additional light on the modulating effect of lactic-fermented bee pollen postbiotics using a human dysbiotic microbiota that corresponds to the pathophysiology of hypertension. To our knowledge, this is the first study to report that FBPP is able to restore a healthy gut microbiota composition, increasing beneficial bacteria and decreasing pathobionts. Moreover, the modulating effect on the microbiota composition was correlated with the metabolic profile after postbiotic treatment. SCFAs like lactate, acetate, and isovaleric acid obtained from pollen fermentation and as microbial metabolites during intestinal simulation contribute to the overall pool of beneficial molecules in the gut. Further studies are needed to assess the phenolic components of FBPP and to understand their cardioprotective role. These preliminary studies highlight the therapeutic potential of FBPP, but disease-induced animal model studies and clinical trials are needed to evaluate the functionality of these preparations. On the other hand, postbiotic products increase the bioavailability of bioactive molecules and exhibit an enhanced safety and practical advantages (longer shelf life and simple formulation protocol, without the loss of activity). Understanding the impact of postbiotic products on normal or dysbiotic gut microbiota could lead to innovative strategies for promoting heart health and preventing cardiovascular diseases. 

## 5. Conclusions

By exploring the potential of pollen-derived postbiotics, we can continue to unlock new possibilities for promoting gut health and overall well-being. This study investigated the impact of lactic-fermented bee pollen postbiotics on the cardiovascular microbiota via in vitro tests. Both liquid and dried postbiotics increased the number of beneficial bacteria (*Lactobacillus* spp. and *Bifidobacterium* spp.), while the amount of opportunistic pathobionts decreased, but they had no significant effect on the *Firmicutes*/*Bacteroidetes* ratio. The modulating effect on the microbial composition was correlated with the metabolic profile, revealing significant increases in lactic and acetic acids, both with known antihypertensive effects, and isovaleric acid, which promotes a healthy gut microbiota.

## Figures and Tables

**Figure 1 biomedicines-11-02712-f001:**
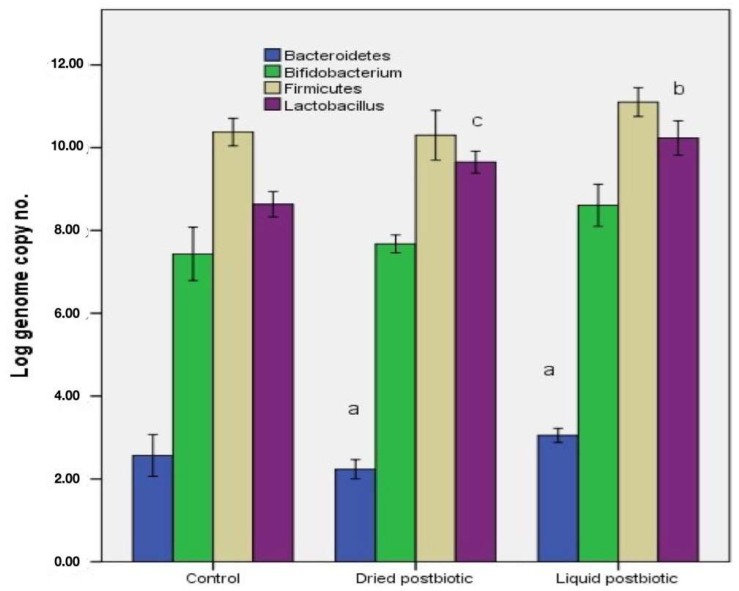
The modulation effect of liquid and dried fermented bee pollen postbiotic on the microbial composition of cardiovascular dysbiosis, determined by the qPCR technique. Different letters represent significant statistical differences (*p* ≤ 0.05) between samples and control, *n* = 3.

**Figure 2 biomedicines-11-02712-f002:**
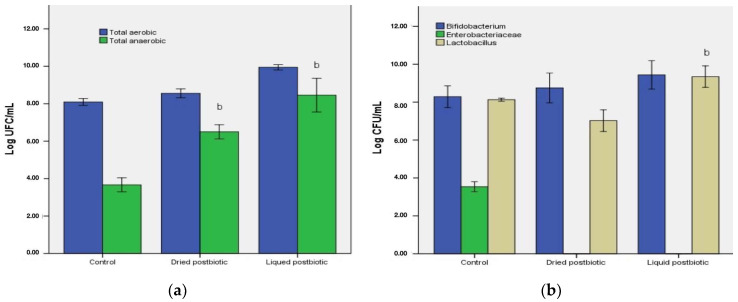
The modulation effect of liquid and dried fermented bee pollen postbiotic on the microbial composition of cardiovascular dysbiosis, determined by a microbiological method. (**a**) Total aerobic and anaerobic counts on non-selective media; (**b**) *Lactobacillus* spp., *Bifidobacterium* spp. and *Enterobacteriaceae* counts on selective media (MRS agar, BSM agar, MacConkey agar). Different letters represent significant statistical differences (*p* ≤ 0.05) between samples and control, *n* = 3.

**Table 1 biomedicines-11-02712-t001:** Organic acid composition of the dried fermented bee pollen postbiotic analyzed by the zonal capillary electrophoresis method.

Organic Acids	mg/mL	Organic Acids	mg/mL
Acetic acid	2.237 ± 0.027	Izovaleric acid	1.519 ± 0.014
Propionic acid	0.051 ± 0.002	Oxalic acid	0.023 ± 0.000
Lactic acid	18.671 ± 0.429	Benzoic acid	-
Butyric acid	1.066 ± 0.016	Formic acid	-
Hydroxyphenyllactic acid	0.005 ± 0.000	Succinic acid	-
Phenyllactic acid	0.008 ± 0.000		

**Table 2 biomedicines-11-02712-t002:** Organic acid composition of the dried and liquid fermented bee pollen postbiotic compared to the control sample, analyzed by the zonal capillary electrophoresis method.

Organic Acids	Control (g/L)	Dried Postbiotic (g/L)	Liquid Postbiotic (g/L)
Acetic acid	3.083 ± 0.030	4.284 ± 0.017	2.872 ± 0.014
Propionic acid	4.844 ± 0.085	0.102 ± 0.012	0.046 ± 0.004
Lactic acid	0.701 ± 0.020	6.386 ± 0.106	5.435 ± 0.131
Butyric acid	-	0.023 ± 0.00	0.077 ± 0.002
Hydroxyphenyllactic acid	-	0.023 ± 0.000	0.021 ± 0.000
Phenyllactic acid	0.048 ± 0.00	0.063 ± 0.002	0.075 ± 0.00
Izovaleric acid	0.544 ± 0.008	2.725 ± 0.037	2.096 ± 0.016
Succinic acid	1.257 ± 0.029	0.223 ± 0.028	0.488 ± 0.017
Benzoic acid	0.061 ± 0.006	-	-
Formic acid	-	-	-
Oxalic acid	-	-	-

## Data Availability

The graphical abstract was created with BioRender.com.

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
