# Peer review of "The Modulation Effect of a Fermented Bee Pollen Postbiotic on Cardiovascular Microbiota and Therapeutic Perspectives"

_biomedicines, 2023, doi:10.3390/biomedicines11102712_

Round 1
Reviewer 1 Report
The present study focused on the impact of the lactic fermented bee pollen postbiotic (FBPP) on cardiovascular microbiota by in vitro tests. The topic of this study fully fills in the scope of Biomedicines. There were several comments for further improving the quality of the manuscript.
1. Keywords, “modulation effect on microbiota composition and metabolic profile” was too long.
2. Section 3.2, the Firmicutes/Bacteroidetes ratio was closely associated with different dietary structure.
3. The labels in Figure 1, such as a, b, and c, should be explained.
4. Figure 2, the quality should be improved. And the labels a and b should also be explained.
5. The present study was a research article. So, figure 3 is not recommended.
Author Response
Thank you for the review of our manuscript. We have revised the manuscript in response to comments by the three reviewers. To follow please find an itemized list of reviewer comments in addition to the manner in which we have addressed the reviewer's concern.
The present study focused on the impact of the lactic fermented bee pollen postbiotic (FBPP) on cardiovascular microbiota by in vitro tests. The topic of this study fully fills in the scope of Biomedicines. There were several comments for further improving the quality of the manuscript.
- Keywords, “modulation effect on microbiota composition and metabolic profile” was too long.
The authors agree with the reviewer and changed to “microbiota modulation effect”
- Section 3.2, the Firmicutes/Bacteroidetes ratio was closely associated with different dietary structure.
The reviewer is correct and some phrases have been added to the text (lines 230-233):
“The two major phyla Firmicutes and Bacteroidetes include almost 90% of microbial species inhabiting the human gut and an increased F/B ratio has been associated with CV diseases, obesity, diabetes, and inflammatory bowel disease, as well as with different dietary structures [6, 26].”
- The labels in Figure 1, such as a, b, and c, should be explained.
The reviewer is correct and some phrases have been added to the figure:
“Different letters represent significant statistical differences (p ≤ 0.05) between samples vs. control, n = 3”
In Material and methods is mention the significance level:
“The significance level for the calculations was set as follows: significant, p<0.05; very significant, p<0.01; highly significant, p<0.001; and extremely significant p<0.0001 using the letters from a to d in text.”
- Figure 2, the quality should be improved. And the labels a and b should also be explained.
The reviewer is correct and some phrases have been added to the figure:
“Different letters represent significant statistical differences (p ≤ 0.05) between samples vs. control, n = 3”
- The present study was a research article. So, figure 3 is not recommended.
The reviewer is correct and Figure 3 was deleted from the text and moved as a graphical abstract.
Reviewer 2 Report
Dinu et al. reported that FBPP are able to restored a healhty gut microbiota composition, increasing beneficial bacteria and decreased pathobionts. Moreover, the modulation effect on the microbiota composition was correlated with the metabolic profile after postbiotic treatments. SCFAs like lactate, acetate and isovaleric acid obtained from pollen fermentation and as microbial metabolites during intestinal simulation contribute to the overall pool of beneficial molecules in the gut. Overall, this article does not have enough scientific soundness.
Main limitations of this work include:
Only in vitro studies were performed. The exact amount of FBPP to be beneficial is unknown.
No functional studies were performed, such as BP examined.
The authors stated 2-way ANOVA were applied, however, I cannot see the 2 main factors described in result section and no interaction was described.
Fig. 3 is speculative, without much evidence support from this study.
The main findings were liquid and dried postbiotic increased the number of beneficial bacteria (Lactobacillus spp. and Bifidobacterium spp.). My feeling was the data look preliminary.
Author Response
Thank you for the review of our manuscript. We have revised the manuscript in response to comments by the three reviewers. To follow please find an itemized list of reviewer comments in addition to the manner in which we have addressed the reviewer's concern.
Dinu et al. reported that FBPP are able to restored a healhty gut microbiota composition, increasing beneficial bacteria and decreased pathobionts. Moreover, the modulation effect on the microbiota composition was correlated with the metabolic profile after postbiotic treatments. SCFAs like lactate, acetate and isovaleric acid obtained from pollen fermentation and as microbial metabolites during intestinal simulation contribute to the overall pool of beneficial molecules in the gut. Overall, this article does not have enough scientific soundness.
Main limitations of this work include:
- Only in vitro studies were performed. The exact amount of FBPP to be beneficial is unknown.
The reviewer has raised a good point but assessing the amount of FBPP with known health effects needs deeper studies about the composition of bioactive molecules from FBPP and their overall effects on human health. The need for more research was mentioned in the text regarding the polyphenols that are in bee bread and are known to have cardioprotective effects (lines 456-457):
“Further studies are needed to assess the phenolic components from FBPP and to understand their cardioprotective role.“
Moreover, the optimal amount of fermented bee pollen postbiotic for health benefits can vary depending on several factors, including an individual's age, weight, overall health, and specific health goals. Additionally, scientific research on FBPP is limited, so there are no established recommended daily intake guidelines.
- The main findings were liquid and dried postbiotic increased the number of beneficial bacteria (Lactobacillus spp. and Bifidobacterium spp.). My feeling was the data look preliminary.
To our knowledge, this is the first time that the effect of fermented bee pollen postbiotic on human microbiota was investigated in order to understand the therapeutic potential of this approach. The study proved that both, liquid and dried postbiotic increased the number of beneficial bacteria (Lactobacillus spp. and Bifidobacterium spp.), as well as the acetic, lactic, and isovaleric acids level, while the opportunistic pathobionts decreased. There are important findings that encourage us to continue the work.
It was mentioned in the text that this is a preliminary study (lines 84-87):
“This pioneering study showed the changes in CV microbiota composition and metabolic profile after fermented bee pollen postbiotic administration and highlighted the therapeutic perspectives of this approach to promote heart health.”
At the same time, it was mentioned that more research is expected to unveil novel functional components and mechanisms in order to validate these beneficial effects (lines 456-457):
“Further studies are needed to assess the phenolic components from FBPP and to understand their cardioprotective role.“
However, the reviewer has raised a good point and some phrases have been added to the text (lines 457-460):
“These preliminary studies highlight the therapeutic potential of FBPP but disease-induced animal models and clinical trials are needed to evaluate the functionality of these preparations.”
- No functional studies were performed, such as BP examined.
The reviewer has raised a good point and we agree that the bee pollen composition depends on the botanical origin of the plant, the soil, and geographical conditions. However, studies showed that bee bread (naturally fermented bee pollen) has a composition that is different compared to bee pollen and is more accessible in bioactive compounds than bee pollen (Aylac, 2023; Aylac, 2021). Therefore, the authors considered that information about the composition of FBPP is relevant to identify their effects, therefore Table 1 shows information about the organic acids composition of the dried fermented bee pollen postbiotic, the main bioactive compounds found in the tested product.
1.Aylanc, V.; Tomás, A.; Russo-Almeida, P.; Falcão, S.I.; Vilas-Boas, M. Assessment of Bioactive Compounds under Simulated Gastrointestinal Digestion of Bee Pollen and Bee Bread: Bioaccessibility and Antioxidant Activity. Antioxidants 2021, 10, 651. https://doi.org/10.3390/antiox10050651
- Aylanc V, Falcão SI, Vilas-Boas M. Bee pollen and bee bread nutritional potential: Chemical composition and macronutrient digestibility under in vitro gastrointestinal system. Food Chem. 2023 Jul 1;413:135597. doi: 10.1016/j.foodchem.2023.135597.
- The authors stated 2-way ANOVA were applied, however, I cannot see the 2 main factors described in result section and no interaction was described.
A two-way ANOVA is used to estimate how the mean of a quantitative variable changes according to the levels of two categorical variables. The quantitative variables represent mean counts of things (e.g., results for Bifidobacterium spp., Lactobacillus spp.) and 2 categorical variables are qPCR results and microbiological analysis results. This type of statistical section was used in the past and it was accepted in the published papers in journals with the same quotation. It presents only essential data relevant to the study.
The text has been changed (lines 187-189):
“The one-way and two-way ANOVA was used for …”
For Lactobacillus spp. counts were clearly differences between qPCR and microbiological results and that was discussed in the text (lines 382-390):
“In our study, higher Lactobacillus spp. density after postbiotic-treated samples was correlated with lactic acid concentration. In samples treated with spray-dried FBPP lactobacilli density determined by cultivation onto MRS media was 7.759±1.187 log CFU/mL and lactate 6.386±0.106 g/L, while for liquid FBPP the lactic acid bacteria population was 9.117±0.525 log CFU/mL and lactate 5.435±0.131 g/L. The higher number of lactic bacteria detected by qPCR in samples treated with FBPP could be linked with the presence of DNA from honeybee Lactobacillus inoculum, but increased amount of culturable lactobacilli onto specific media compared to control is a real effect of liquid postbiotic.”
- Figure 3 is speculative, without much evidence support from this study.
The reviewer is correct and Figure 3 was deleted from the text and moved as a graphical abstract.
Reviewer 3 Report
In this report the authors prepared a fermented bee pollen postibiotic (FBPP) with a newly isolate Lactobacillus spp which produced (by anaerobic fermentation) a specific composition of organic acids rich in acetic and lactic acids (table 1) as liquid solution and dried powder after thermal treatment. The FBBPs were used by the investigators in a single chamber in vitro simulation of the human gastrointestinal tract to a microbiome content related to cardiovascular disease, dyslipidemia and hypertention for 7-10 days. The microbiome from the simulated system was analyzed by microbiological methods and qPCR. The authors found that treatment with FBBP increased the content of beneficial bacteria, checking Firmicutes/Bacteroidetes ratios, whilst the increased lactic and acetic acid, and the presence of isovaleric promote healty anti-cardiovascular disease or hypertention gut microbiome.
Concerns
1. The authors should provide more about their experimental system the exact steps of its configuration in the preparation of samples for microbiome examination.
2. This is an in vitro study based on a gastrointestinal simulator controlled for pH and temperature in a biphasic operational mode (simulating stomach and small intestine in phase 1 and then colon in phase2) with human fecal samples but there is no information for the clinical status of the patients involved. This information is critical for the scope of this article. The authors should provide the clinical data of the people provided the samples (feces) tested in the GIS1 system.
3. The samples (feces) of the patients tested had already a microbiota composition. It is important to report the changes of the feces composition before and after the treatment with the FBPP.
4. The evidence presented in figures 1 and 2 are not convincing for the detection of deferences between controls, dried, and liquid postbiotic. The deviations presented are too high compared to the mean values. There is no mention of the number of samples used in any analysis.
Author Response
Thank you for the review of our manuscript. We have revised the manuscript in response to comments by the three reviewers. To follow please find an itemized list of reviewer comments in addition to the manner in which we have addressed the reviewer’s concern.
In this report the authors prepared a fermented bee pollen postibiotic (FBPP) with a newly isolate Lactobacillus spp which produced (by anaerobic fermentation) a specific composition of organic acids rich in acetic and lactic acids (table 1) as liquid solution and dried powder after thermal treatment. The FBBPs were used by the investigators in a single chamber in vitro simulation of the human gastrointestinal tract to a microbiome content related to cardiovascular disease, dyslipidemia and hypertention for 7-10 days. The microbiome from the simulated system was analyzed by microbiological methods and qPCR. The authors found that treatment with FBBP increased the content of beneficial bacteria, checking Firmicutes/Bacteroidetes ratios, whilst the increased lactic and acetic acid, and the presence of isovaleric promote healty anti-cardiovascular disease or hypertention gut microbiome.
Concerns
1. The authors should provide more about their experimental system the exact steps of its configuration in the preparation of samples for microbiome examination.
The reviewer has raised a good point but based on our experience with MDPI journals the recommendation was to avoid repetition of the same data in different papers. In this case, reference #19 contains additional data about the sample preparation. Similarly, the in vitro GIS system was already presented in different articles (references # 24, #25).
19.Gatea, ; Sârbu, I.; Vamanu, E. In Vitro Modulatory Effect of Stevioside, as a Partial Sugar Replacer in Sweeteners, on Human Child Microbiota. 2021,Microorganisms, 9, 590.
24.Vamanu E, Pelinescu D, Sarbu I. Comparative Fingerprinting of the Human Microbiota in Diabetes and Cardiovascular Disease. J Med Food. 2016, 19(12), 1188-1195. doi: 10.1089/jmf.2016.0085.
25.Vamanu E, Gatea F, Sârbu I, Pelinescu D. An In Vitro Study of the Influence of Curcuma longaExtracts on the Microbiota Modulation Process, In Patients with Hypertension. 2019, Pharmaceutics, 11(4), doi: 10.3390/pharmaceutics11040191.
2.This is an in vitro study based on a gastrointestinal simulator controlled for pH and temperature in a biphasic operational mode (simulating stomach and small intestine in phase 1 and then colon in phase2) with human fecal samples but there is no information for the clinical status of the patients involved. This information is critical for the scope of this article. The authors should provide the clinical data of the people provided the samples (feces) tested in the GIS1 system.
The reviewer is correct and the text has been changed to (lines 118-136):
“In vitro simulation using the GIS1 system
Tests were conducted using a gastrointestinal simulator - GIS1 system, only phase 2 - human colon (www.gissystems.ro) and the microbiome of the target CV disease—dyslipidemia and hypertension group. The reconstitution process followed the protocol previously described and was performed with a mean interval of 7–10 days [19]. The samples (feces) were handled by the ethical guidelines of UASVM Bucharest (ColHumB Registration number: 1418/23.11.2017; www.colhumb.com) and analyzed individually. The biological samples were collected in 10% glycerol and stored at −15 °C until needed. All samples are from persons clinically diagnosed with cardiovascular pathologies (especially hypertension), and they did not take medications that could influence the microbiota pattern (for example, antibiotics). The liquid and capsules with FBPP were directly added in the simulated environment under sterile conditions. The capsules used were gastro-resistant to pass phase 1 of the simulation (stomach and small intestine). pH evolution was recorded during the simulation process. At the end of the in vitro simulation (day 10), each sample collected was subjected to centrifugation 4000×g for 15 min (Hettich Universal 320, Hettich GmbH., Germany), and the sediment (microbial fingerprint) was microbiologically analyzed within 24h or preserved in glycerol 20% for qPCR analysis. The supernatant was stored in a refrigerator at −15 °C for chemical analysis.”
The gastro-resistant capsules ensure that all bioactive compounds from the capsules pass into the colon and modulate the microbiota pattern and bioactivity.
- The samples (feces) of the patients tested had already a microbiota composition. It is important to report the changes of the feces composition before and after the treatment with the FBPP.
The authors considered that the negative control is the sample named “control” that shows the microbiota composition without the FBPP treatment. The microbiota used was previously analyzed and corresponds to the pathophysiology of hypertension. Therefore, the authors considered it more relevant to compare results at the end of the simulations (performed in the same conditions) with/without FBPP treatment.
4. The evidence presented in figures 1 and 2 are not convincing for the detection of deferences between controls, dried, and liquid postbiotic. The deviations presented are too high compared to the mean values. There is no mention of the number of samples used in any analysis.
The reviewer is correct and we checked the data and discovered some mistakes. Therefore, Figures 1 and 2 were redrawn.
The number of samples was mentioned in Materials and Methods (lines 185-186):
“All the parameters were evaluated in triplicate and the results were expressed as mean standard deviation (SD).”
Also, for qPCR analysis was mentioned (lines 161-162):
“Reactions were carried out in triplicate, and the results were statistically analyzed.”
The text has been changed (line 136):
“Two different experiments were performed.”
Round 2
Reviewer 1 Report
The quality of the revised manuscript was improved.
Author Response
Thank you for the review of our manuscript.
Reviewer 2 Report
This study does not carry enough scientific impact.
Author Response
Cardiovascular diseases (CVD) are the leading cause of death in the world and a recent report from the European Society of Cardiology estimated that CVD cost the EU €282 billion in 2021 (https://www.escardio.org/The-ESC/Press-Office/Press-releases/Price-tag-on-cardiovascular-disease-in-Europe-higher-than-entire-EU-budget).
Cardiovascular health promotion is based on promoting and maintaining low cardiovascular risk, including the risk associated with gut dysbiosis. Therefore, finding new approaches based on gut microbiota modulation for cardiovascular prevention, especially for high-risk individuals or to ameliorate the symptoms associated with CVD is important. That was mentioned in the manuscript.
Lines 52-54: “Targeting gut microbiota to prevent and treat hypertension or to ameliorate the symptoms associated with different CV pathology is a feasible direction for further adjuvant therapies.”
In this work, for the first time was investigated the effect of fermented bee pollen postbiotic (FBPP) on human CV microbiota in order to understand the therapeutic potential of this approach. Also, it was mentioned that the study is a preliminary research that will enhance the interest toward new products based on FBPP.
Lines 84-87 “This pioneering study showed the changes in CV microbiota composition and metabolic profile after fermented bee pollen postbiotic administration and highlighted the therapeutic perspectives of this approach to promote heart health.“
Reviewer 3 Report
In this report the authors prepared a fermented bee pollen postibiotic (FBPP) with a newly isolate Lactobacillus spp which produced (by anaerobic fermentation) and test it in an in vitro system simulating the human gastrointestinal system with feces from cardiovascular diseased patients. The authors used as control untreated feces of the same origin. It is not known what are the effects of FBPP in feces from apparent healthy (not with a cardiovascular disease aetiology) people. The microbiota composition as well
as the organic acids composition should be assessed before and after FBPP treatment.
The exact clinical condition of the patient donors of feces is not clarified. Significant
differences could be anticipated for the microbiota composition of different pathologies
myocardial infarction, dyslipidemia, diabetes or hypertention.
Author Response
1.The authors used as control untreated feces of the same origin. It is not known what are the effects of FBPP in feces from apparent healthy (not with a cardiovascular disease aetiology) people.
Investigating the effects of FBPP on gut microbiota using feces from healthy individuals may be an avenue for future research. These studies could provide valuable insights into whether FBPP has any discernible effects on the gut microbiome or other relevant factors in individuals without cardiovascular disease, shedding light on its broader potential applications and safety profile.
The authors consider that preliminary results obtained in our study will enhance the interest in new products based on FBPP which can be used in different therapy or to promote well-being.
2.The microbiota composition as well as the organic acids composition should be assessed before and after FBPP treatment.
It was mentioned in the manuscript that the CV intestinal microbiota composition before FBPP treatment was previously analyzed.
Lines 231-234 “The CV gut microbiota used in the study has been previously analyzed and used in different studies [24, 25]. This microbiota corresponds to the pathophysiology of hypertension…”
The organic acids composition before FBPP treatment was analyzed and is similar to the composition determined in previous research (Vamanu, 2019).
Vamanu, E.; Gatea, F.; Sârbu, I.; Pelinescu, D. An In Vitro Study of the Influence of Curcuma longa Extracts on the Microbiota Modulation Process, In Patients with Hypertension. Pharmaceutics 2019, 11, 191. https://doi.org/10.3390/pharmaceutics11040191
Therefore, the text has been changed to:
Lines 287-288 “Before FBPP treatment small amounts of lactic acid (0.39±0.05) g/L), acetic acid (0.34±0.04 g/L) and propionic acid (0.018±0.01) were detected, similar to previously published data [28].”
References have been changed.
3.The exact clinical condition of the patient donors of feces is not clarified. Significant differences could be anticipated for the microbiota composition of different pathologies myocardial infarction, dyslipidemia, diabetes or hypertention.
The relevant information was included in the designated section of the Materials and Methods.
Lines 118-137:
“In vitro simulation using the GIS1 system
Tests were conducted using a gastrointestinal simulator - GIS1 system, only phase 2 - human colon (http://www.gissystems.ro/) and the microbiome of the target CV disease—dyslipidemia and hypertension group. The reconstitution process followed the protocol previously described and was performed with a mean interval of 7–10 days [19]. The samples (feces) were handled by the ethical guidelines of UASVM Bucharest (ColHumB Registration number: 1418/23.11.2017; www.colhumb.com) and analyzed individually. The biological samples were collected in 10% glycerol and stored at −15 °C until needed. The samples used in this study were obtained from individuals clinically diagnosed with cardiovascular diseases, including hypertension and dyslipidemia. It is important to note that these individuals did not use any medications that could potentially impact the pattern of their microbiota, such as antibiotics. The collecting techniques for fecal samples adhered to the ethical guidelines set forth by the University of Agricultural Sciences and Veterinary Medicine in Bucharest, Romania. Three individuals (two men and one female) aged 45 and 70 were selected to collect fecal samples. The frozen glycerol samples were reconstituted in phosphate-buffered saline (PBS) to achieve a consistent microbial profile. The liquid and capsules with FBPP were directly added to the simulated environment under sterile conditions. The capsules used were gastro-resistant to pass phase 1 of the simulation (stomach and small intestine). pH evolution was recorded during the simulation process. “
Round 3
Reviewer 2 Report
A third reviewer is advised.
